# Effects of Different Ratios of Carbohydrate–Fat in Enteral Nutrition on Metabolic Pattern and Organ Damage in Burned Rats

**DOI:** 10.3390/nu14173653

**Published:** 2022-09-04

**Authors:** Yongjun Yang, Sen Su, Yong Zhang, Dan Wu, Chao Wang, Yan Wei, Xi Peng

**Affiliations:** 1Clinical Medical Research Center, Southwest Hospital, Third Military Medical University (Army Medical University), Chongqing 400038, China; 2State Key Laboratory of Trauma, Burns and Combined Injury, Institute of Burn Research, Southwest Hospital, Third Military Medical University (Army Medical University), Chongqing 400038, China; 3Shriners Burns Hospital, Massachusetts General Hospital, Harvard Medical School, Boston, MA 02114, USA

**Keywords:** burns, carbohydrate–fat ratio, enteral nutrition, hypermetabolism, organ damage

## Abstract

(1) Background: Nutritional support is one of the most important cornerstones in the management of patients with severe burns, but the carbohydrate-to-fat ratios in burn nutrition therapy remain highly controversial. In this study, we aimed to discuss the effects of different ratios of carbohydrate–fat through enteral nutrition on the metabolic changes and organ damage in burned rats. (2) Methods: Twenty-four burned rats were randomly divided into 5%, 10%, 20% and 30% fat nutritional groups. REE and body weight were measured individually for each rat daily. Then, 75% of REE was given in the first week after burns, and the full dose was given in the second week. Glucose tolerance of the rats was measured on days 1, 3, 7, 10 and 14. Blood biochemistry analysis and organ damage analysis were performed after 7 and 14 days of nutritional therapy, and nuclear magnetic resonance (NMR) and insulin content analysis were performed after 14 days. (3) Results: NMR spectra showed significant differences of glucose, lipid and amino acid metabolic pathways. The energy expenditure increased, and body weight decreased significantly after burn injury, with larger change in the 20%, 5% and 30% fat groups, and minimal change in the 10% fat group. The obvious changes in the level of plasma protein, glucose, lipids and insulin, as well as the organ damage, were in the 30%, 20% and 5% fat groups. In relative terms, the 10% fat group showed the least variation and was closest to normal group. (4) Conclusion: Lower fat intake is beneficial to maintaining metabolic stability and lessening organ damage after burns, but percentage of fat supply should not be less than 10% in burned rats.

## 1. Introduction

The intense stress and ongoing inflammatory response after major burns puts the body in a state of hypermetabolism for a long time, which is a key cause for increased energy expenditure, loss of lean tissue, immunosuppression, and consequently systemic infection, organ damage and wound healing delay [1,2,3,4,5]. Nutritional support represents one of the most important cornerstones in the management of patients with severe burns, especially to cope with hypermetabolism [1,6,7]. There are many important issues involved in nutrition therapy for burns, including the total calorie supply, nutritional pathway and nutrient ratios. Notably, the principles of macronutrient rationing remains highly controversial [8,9].

Carbohydrates, fats and proteins are nutrients needed by the human body. The appropriate ratio among them is particularly important for burn patients’ nutrition, and may result in better patient outcomes [8,10]. There is a consensus on the amount of protein to be supplied to burn patients, which is generally at 15–20% of total calories, or 1.5–2.0 g/kg [11,12]. However, the ratio of carbohydrate to fat in nonprotein calories remains highly controversial. Several burn nutrition guidelines and relevant monographs have made recommendations for fat calorie ratios, which are approximately 20–30% [2,6,13]. However, its actual ratio in the literature varies significantly from 2% to 56% [8]. The ratio is related to the patient’s degree of damage, age and course of disease, and also to the nutritional protocol of different medical units, which is influenced by the constantly updated knowledge of nutrients. From the 1980s to the early 2000s, the fat supply in burn patients was typically 30–50% or even higher [14,15,16]. At that time, the two main advantages of fat were based on its high caloric density and low CO_2_ production [17,18]; however, not enough attention was given to whether the body could tolerate fat well after burns. Subsequently, it was found that excessive fat intake could lead to problems such as hyperlipidemia, fatty deposits in the liver, and immunosuppression [5,16,19,20]. In recent years, the fat supply has been gradually reduced, and 15–30% fat is generally used for energy supply [21,22].

In recent years, the nitrogen-saving effects of glucose have become increasingly valued, and low-fat, high-carbohydrate formulations have gradually become the main protocol in burn nutrition [2,8]. Despite the obvious advantages of low-fat formulations, it is debatable whether a very low-fat supply is better [8,9,16,23]. Some nutritional formulas of burn units have only 2–4% of the total calories supplied by fat [17,24], which inevitably leads to excessive carbohydrate intake, which is unhelpful for controlling post-burn hyperglycemia. Furthermore, such a low-fat intake fails to meet the body’s requirements for essential fatty acids and will lead to poor outcomes [25,26].

Up to now, the majority of clinical tests exploring the best nonprotein calorie ratios have only two groups: a high-carbohydrate low-fat group and a low-carbohydrate high-fat group [22,27,28]. In some trials, the fat intake of burned patients varied by a factor of more than 10, which makes it difficult to screen out optimal carbohydrate–fat ratios [8,29,30]. Based on the ethical principles and the reality of limited clinical cases of burn patients, this study used an enteral nutrition model in burned rats. On the basis of uniform calorie and protein supply, the burned animals were divided into four groups according to the amount of fat supplementation from 5% to 30%, in order to observe the effects of different fat supplies on the metabolic changes and organ damage after burns. Finally, we aimed to find the optimal carbohydrate–fat ratio and provide an experimental basis for optimizing the nutritional formula for burn patients.

## 2. Materials and Methods

### 2.1. Experimental Animals

Male Sprague Dawley (SD) rats (specific pathogen-free, SPF), 6–8 weeks old, weighing 250 ± 10 g, were purchased from the Animal Experiment Center of Daping Hospital, Third Military Medical University, China. The rats were housed and maintained in the SPF-class animal facilities of the Clinical Medicine Research Center of Southwest Hospital, Third Military Medical University. The rats were given an ad libitum diet and water for one week prior to the experiments, and the feeding room temperature was 25 °C with 40–60% relative humidity and alternating light/darkness every 12 h. Food and water were abstained for 12 h prior to scalding. Third Military Medical University’s Institutional Animal Ethics Committee approved all animal experiments according to the National Animal Welfare Guidelines (Approved Agreement Number AMUWEC2020014).

### 2.2. Preparation of the Animal Burn Model and Nutritional Treatment Regimens

#### 2.2.1. Burn Model and Grouping

Forty-eight rats were divided into 8 groups (4 groups each for burns and normal control) by the randomized numerical table method. The rats were given enteral nutrition and the same proportion of calories and protein according to their energy consumption. The burned and normal rats were divided into 4 nutritional pattern groups based on the different ratios of carbohydrate and fat intake. As an analgesic, pentobarbital 40 mg/kg and buprenorphine (1 mg/kg body weight) were given to all rats. The dorsum was shaved, weighed, and the shaved area was scalded with hot water (95 °C, 15 s) to cause III° burns on 20% of the body surface area. After the burn, fluid resuscitation was performed by intraperitoneal injection of lactated Ringer’s solution at 40 mL/kg. Each animal was placed separately in a single cage and kept warm for 72 h in a burn holding frame. The traumatic surface was coated with iodophor anti-infection twice a day. The rats in the normal group were immersed in 37 °C warm water for 15 s. Anesthesia was administered but resuscitation was not performed.

#### 2.2.2. Nutritional Treatment Regimens

For daily energy supply, each animal was determined based on the actual measurement value of resting energy expenditure (REE), as described in Section 2.3.2. According to the tolerance of burned rats, 75% of REE was given in the first week after burns, and the full dose was given in the second week. The design bytes of the whole experiment are in Figure 1. For daily diet, based on Peptisorb (Nutricia Pharmaceutical Co., Ltd., Wuxi, China), the fat calorie supply ratio was adjusted to 5%, 10%, 20% and 30% by adding medical starch, amino acid powder or medium/long chain fatty acid injection. The ratio of protein was fixed at 20%, and the final nutritional preparation was divided into 4 dosage forms with different carbohydrate-fat ratios, as detailed in Appendix A. Nutrition was provided by gavage post-burn. The ratios of each group are shown below: ① Burn + 5% fat group (5% group), 75% carbohydrate: 5% fat: 20% protein; ② Burn + 10% fat group (10% group), 70% carbohydrate: 10% fat: 20% protein; ③ Burn + 20% fat group (20% group), 60% carbohydrate: 20% fat: 20% protein; ④ Burn + 30% fat group (30% group), 50% carbohydrate: 30% fat: 20% protein. Correspondingly, each burned group was paired with a normal group, using the same nutritional pattern. Daily REE measurements and actual energy supply data are shown in Appendix A.

### 2.3. Test Indicators

#### 2.3.1. Body Weight Measurement

Individual animals were weighed individually at 8 am each day using an electronic balance (Sartorius, Göttingen, Germany) with an accuracy of 1/1000, and these values were recorded.

#### 2.3.2. REE Measurement

CO_2_ and O_2_ concentrations were measured using Columbus Instruments (OH) for indirect calorimetry to determine the REE. The rats were placed in a Plexiglas metabolic chamber (4 L volume). Calcium sulfate columns were placed at the air inlet and outlet to ensure air dryness. During six cycles (lasting 60 min), the airflow rate was continuously monitored for 10 min, and the amount of O_2_ consumed and CO_2_ produced was calculated by multiplying the airflow rate by the difference between inhaled and exhaled concentrations of O_2_. CO_2_ (DCO_2_) output and REE were calculated using Oxymax software (Columbia Instruments, Columbus, OH, USA) based on the difference in O_2_ intake (DO_2_). The REE was calculated before burn and on post-burn days 1–14 using the following equation: REE (Kcal) = [3.94 × DO_2_ (L/min) + 1.1 × DCO_2_ (L/min)] × 1440 × 4.184.

#### 2.3.3. Glucose Tolerance Assay

Three rats were randomly selected from each group for the glucose tolerance assay on days 1, 3, 7, 10 and 14 post-burn. The rats were fasted for 12 h before the test, and fasting blood glucose was measured by the tail-cutting method. A 10% glucose solution was injected intraperitoneally (2 g/kg according to the body weight of the rats) after 10 min, and the blood glucose values were measured at 10, 30, 60, 90 and 120 min after glucose injection; the unit of blood glucose recording was mmol/L. The recorded data were plotted as a blood glucose concentration–time change curve.

#### 2.3.4. Blood Biochemistry Test Indicators

On days 7 and 14 post-burn, 5 and 6 mL of blood was drawn from the abdominal aorta after anesthesia and analgesia, respectively. On the 14th day, 1 mL of the sample was anticoagulated with sodium citrate tubes, and the remaining 5 mL of whole blood was placed in EDTA procoagulant tubes. It was centrifuged at 4 °C 3000 rpm for 10 min after being kept at room temperature for 30 min, and the plasma was taken and stored at −80 °C for concentrated testing of liver and kidney function, cardiac enzyme profile, lipids, protein content and insulin. Test indicators included: Kidney damage: blood urea nitrogen (BUN), creatinine (CrEAT), uric acid (UA); *Evaluation of myocardial cell injury*: lactate dehydrogenase (LDH), alpha-hydroxybutyrate dehydrogenase (alpha-HBDH), creatine kinase isoenzyme (CK-MB); Liver damage: glutamate aminotransferase (AST), glutamic aminotransferase (ALT), glutamate transpeptidase (GGT), alkaline phosphatase (ALP), total bilirubin (TBIL), total bile acids (TBA); Blood lipids and protein levels: triglycerides (TG), total cholesterol (Tch), high-density lipoprotein cholesterol (HDL-C), albumin (AlB), low-density lipoprotein cholesterol (LDL-C), total protein (TP). Insulin level: The measurement of insulin level was performed using a radioisotope in vitro microanalysis method that used isotope-labeled and unlabeled antigens to react with antibodies for competitive inhibition. Briefly, the procedure was performed using the commercial [^125^I]-insulin kits explicitly, followed by the determination of the radioactive count (cpm) of the precipitate obtained from each reaction tube using a gamma counter, and finally the data were processed using log-logit software and the insulin levels were obtained.

### 2.4. Metabolic Testing

To observe the changes in body metabolism after burns with different carbohydrate–fat ratios, we constructed a severe burn rat model and gave rats different carbohydrate–fat ratios to observe the differences in body metabolism. After 14 days of burn, the sodium citrate anticoagulated plasma described above (2.3.4) was centrifuged at 4 °C for 10 min at 16,000 rpm, and then 450 μL of plasma was placed in an NMR tube. In the next step, 50 μL of deuterium oxide (D_2_O) were added and shaken thoroughly for 120 s. Next, a 600-MHz NMR (Bruker Biospin, DRX, Billerica, MA, USA) unit was used for measurements and analysis of samples after they were allowed to stand for 10 min.

### 2.5. NMR Spectrum Data Analysis

Plasma ^1^H-NMR data were imported into MestReNova 12.0.1 software (Mestrelab Research, Santiago de Compostela, Galicia, Spain) for analysis. A Fourier transformation was used to transform the free induction decays (FIDs), and the spectra results were phased and baseline-corrected before the chemical shifts were determined. The chemical shifts of the plasma metabolite spectra were determined by referring to the methyl resonance peak of lactate at δ 1.32. The region of chemical shifts between 0 and 8 ppm was subdivided into 2000 intervals with a width of 0.004 ppm. Absorption spectra of water with chemical shifts between 4.7 and 5.1 ppm were removed [31]. Finally, the data were normalized to eliminate dilution-, volume-, or mass-related differences between samples. The same total integration value was assigned to each spectrum before analysis. The characteristic differences of plasma 1H NMR spectra of different groups were compared by multivariate statistical analysis. Based on SIMCA-P software (version 14.1, Umetrics, Ume, Sweden), orthogonal partial least squares regression models have been developed. The supervised model OPLS was evaluated by the goodness of fit (*R*^2^ Y) and goodness of prediction (*Q*^2^), along with the parameters determined by the permutation test (performed using 200 permutation) [32]. The characteristics of metabolites with intergroup differences were determined based on the S-plot curve of the OPLS model and the score for variable importance in projection (VIP > 1) [33]. Substance identification of the screened metabolite features in the original spectra was performed by using Chenomx NMR 8.5 (Chenomx, Edmonton, Canada) and the Human Metabolome Database (HMDB) [34,35]. The screened metabolites were entered into MetaboAnalyst 5.0 for pathway analysis to identify influential pathways [36].

### 2.6. Statistical Analysis

Shapiro–Wilk tests were performed to determine if continuous variables were normally distributed. All data were compounded with a normal distribution, and the means of normally distributed continuous variables were reported as means ± standard deviation (SD). The two groups were compared by using an independent samples *t*-test. Data were analyzed using one-way ANOVA and the Bonferroni test for multiple comparisons between multiple groups. The changes in the repeated measures were statistically analyzed using two-way repeated measures ANOVA and Bonferroni’s test for multiple comparisons. The SPSS program (SPSS 25.0, GraphPad 8.0 and R, version 4.0, SPSS Inc., Chicago, IL, USA) was used for all statistical analyses, and statistical significance was considered if the *p*-value < 0.05.

## 3. Experimental Results

### 3.1. Effects of Different Ratios of Carbohydrate–Fat on Metabolic Patterns

To investigate metabolism changes after burns in the presence of different fat formulations nutritionally, plasma NMR assays and multivariate statistical analyses were performed. The representative ^1^H-NMR spectra of the extract of plasma is shown in Figure 1A and Appendix A. There were variations in the peak signal intensities among the samples, as seen in the spectra. Afterwards, metabolite signals were assigned using the previous studies [37,38], the Human Metabolome Database (HMDB), and Chenomx NMR 8.5’s library by comparing their 1H-NMR signals to those of the reference compounds. In this study, 30 metabolites were identified, as shown in Figure 1A and Appendix A. All these metabolites have been previously reported in many studies [39,40,41]. These metabolites include fatty acids, amino acids, organic acids and carbohydrates. To determine the differential metabolites directly related to carbohydrate–fat ratios, we performed OPLS by using the fat percentage as the Y-matrix. In the OPLS score plots (Figure 1B), the plasma extracts of the 5%, 10%, 20% and 30% groups exhibited clear separation with satisfactory goodness of fit (*R*^2^ Y= 0.995, *Q*^2^ = 0.759). From the 200 permutations test, the model showed Y-axis intercepts of *Q*^2^ less than 0.5 (−1.04), indicating that the models are valid and did not show overfitting (Figure 1C). S-plots represent covariance and association loading diagnostics for OPLS models, which provide an overview of the affecting variables on the model and indicate significant metabolites (Figure 1D). The significant differential metabolites were identified from different groups by using an S-plot curve with significant VIP values > 1 and *p*-values < 0.05 (Appendix A). The final metabolites screened in relation to fat ratios were lipid, isoleucine, leucine, 3-hydroxybutyrate, lactate, alanine, pyruvate, dimethylamine, betaine, glycine, oxalic acid (Figure 1E).

In order to systematically identify the pathways that are most prominent in these groups, metabolic pathway analysis (MetPA) was conducted using MetaboAnalyst. An appropriate tool for assessing metabolite significance is the pathway impact factor. From the results, 20 metabolic pathways were identified. Based on the set standards for a pathway impact value > 0.1 and *p* value < 0.05, five metabolic pathways were determined as important metabolic pathways related to the carbohydrate–fat ratios (Appendix A). The results of the analysis showed significant changes in these five pathways: alanine, aspartate, and glutamate metabolism; glyoxylate and dicarboxylic acid metabolism; glycine, serine and threonine metabolism; pyruvate metabolism; and glycolysis/gluconeogenesis (Appendix A, Figure 1F).

### 3.2. Effect of Different Carbohydrate–Fat Ratios on Energy Consumption and Body Weight Loss

The experimental results showed that the REE of burned rats tended to increase, with the largest increase in the 30% group, followed by the 20%, 5% and 10% groups (Figure 2A). The difference was statistically significant in the 10% group compared to the 5%, 20% and 30% groups 8–14 days after the burn (Figure 2A). In addition, the body weight of rats showed a tendency to decrease after burn injury, and the effect of enteral nutrition with different carbohydrate–fat ratios on rat body weight was different. The 30% group showed the most significant decrease in body weight, with a decrease of approximately 20%, followed by the 30%, 20% and 10% fat groups, with a decrease of approximately 18%, 10% and 5% (Figure 2B). These findings showed that a fat ratio of 10% could not only moderately reduce REE after burn injury, but also effectively maintain the body weight of rats, which was a better carbohydrate–fat rationing pattern.

### 3.3. Effects of Different Carbohydrate–Fat Ratios on Glucose Tolerance and Insulin Levels in Rats

The experimental results showed that the change in glucose tolerance was not obvious 1 day after burn injury (Figure 3A), and after 3, 7, 10, and 14 days of burn injury, the 30 min blood glucose of the burn group could not return to normal levels, which indicated that the body’s ability to metabolize glucose decreased after burn injury. There were some differences between the different fat groups, and the 10% and 30% groups could induce rapid recovery of blood glucose at 3 and 7 days post-burn, which was remarkably different from the 5% and 20% groups (Figure 3B,C). The 10% group at 10 and 14 days post-burn induced rapid recovery of blood glucose, which was significantly different from the 5%, 20% and 30% groups (Figure 3D,E), indicating that 10% fat was better for maintaining blood glucose metabolism post-burn. In addition, different carbohydrate–fat ratios could significantly affect insulin levels, which in turn can regulate blood glucose. After 14 days of burn injury, insulin levels were significantly lower in the 5% vs. 10% group than in the 20% and 30% groups (Figure 3F). Overall, the fat ratio of 10% had the least effect on glucose tolerance and insulin, and was a more appropriate carbohydrate–fat rationing pattern.

### 3.4. Effects of Different Carbohydrate–Fat Ratios on Blood Lipid and Protein Levels in Plasma

The plasma levels of Tch, TG, HDL-C, LDL-C, AlB, and TP were remarkably lower in rats after burn injury than in normal controls, suggesting that severe burn injury can cause abnormalities in lipid and protein metabolism (Figure 4A–F). Different fat ratios could affect lipid and protein metabolism in burned rats to different degrees. At 7 days post-burn, LDL-C and TP levels were significantly higher in the 10% group than in the 5%, 20% and 30% groups. At 14 days post-burn, the Tch, LDL-C, HDL-C, and AIB levels in the 10% group were significantly higher than those in the 5%, 20% and 30% groups. Compared to the four groups, the 10% group did the best job in stabilizing lipid and protein levels, the 20% group was the next best, and the 5% and 30% groups were the worst (Figure 4A–F). The results showed that a fat ratio of 10% had better effects on stabilizing blood lipids and protein metabolism.

### 3.5. Effects of Different Carbohydrate–Fat Ratios on the Degree of Organ Damage in Burned Rats

The experimental results showed that organ damage in rats after burn injury was obvious, and the indices reflecting kidney, myocardium, and liver damage were significantly higher than those in the normal group. The effect of different nutritional ratios on organ damage in burned rats was different. The 10% group had remarkably reduced BUN, CrEAT and UA (Figure 5A–C) levels, and also LDH, α-HBDH, CK and CK-MB (Figure 5D–F) levels, reflecting kidney damage and myocardial cell injury, and decreased AST, ALT, GGT, ALP, TBIL and TBA (Figure 5G–L) levels, reflecting liver damage, with obviously better results than the 20%, 5% and 30% groups. Comparison of the four groups showed that the 10% group had the least organ damage, followed by the 20%, 5% and 30% groups. These results suggest that a fat ratio of 10% is effective in reducing the degree of organ damage at 7 days post-burn, and that these effects are more pronounced at 14 days post-burn.

## 4. Discussion

Significant increase in the intense stress and ongoing inflammatory response following severe burns is a key cause of systemic infection, organ damage and wound healing. Nutritional supplementation represents one of the cornerstones of supportive care in managing patients with severe burns. However, the ratio of carbohydrates to fats in burn nutrition therapy is still controversial. A model of enteral nutrition for burned rats was used in this study to observe how different energy sources affected body metabolism, organ damage and prognosis. According to the REE and rats’ tolerance abilities, the burned rats were given energy at 75% and 100% of the REE in the first and second weeks, respectively, and protein at 20% of the total calories. On this basis, burned animals were divided into four groups according to the proportion of carbohydrate–fat in the nonprotein calories, and the proportion of fat in the total calories was 5%, 10%, 20% and 30%. The results showed that the nutritional formulas using 10% fat were significantly better than the other three groups in reducing hypercatabolism after burns, promoting insulin secretion and reducing organ damage, and the optimal nutritional formulas had the following ratios of protein, carbohydrate and fat: 20%: 70%: 10%.

Following burn injuries, metabolic patterns can be significantly affected by different nutritional formulas. Thirty metabolites involving five metabolic pathways were identified from different nutritional formulas groups, including glucose, protein and fat metabolism, etc. In addition to differences in glucose and fat metabolism due to different intakes, amino acid metabolism was also changed. In the four nutritional formulas groups, plasma amino acid levels were different between low-fat ratios groups (5% and 10%) and high-fat ratios groups (20% and 30%), with glutamine, alanine and glycine much higher, and leucine and isoleucine significantly lower. A clinical study found that plasma glutamine, alanine and glycine concentrations were lower after burn injury, leucine and isoleucine were markedly higher, and the magnitude of change had a positive correlation with the degree of burn injury [42,43]. The results of this study suggested that nutritional support with low-fat and high-carbohydrate levels is beneficial in maintaining plasma amino acid balance. Plasma amino acid levels after burns are controlled by multiple factors, including protein intake, protein anabolism and catabolism [44]. When catabolism is excessive and prolonged, the body’s energy reserves are depleted, which results in poor outcomes [45,46]. The protein intake of the four groups of burned rats in this study was consistent; thus, the differences in plasma amino acid levels were mainly regulated by protein synthesis in the liver and protein degradation in skeletal muscle. Combined with the changing trends in REE and body weight of burned rats, the degree of hypermetabolism and skeletal muscle loss in the 5% and 10% groups were notably smaller than the other two groups, suggesting that the supply of higher carbohydrate could reduce body consumption and maintain the content of lean tissue, which is consistent with the nitrogen-saving effect of glucose reported in the literature [47,48,49]. The mechanism may be related to the fact that a higher carbohydrate supply can promote the conversion of glucose to amino acids [31,50]. Our results showed that pyruvate levels were noticeably higher in the 5% and 10% fat groups than in the other two groups, which may be the reason for the high levels of alanine and glutamine. In the liver, pyruvate can produce some amino acids, such as alanine and glutamine, through transamination [51], but cannot promote the transformation to branched chain amino acids (leucine and isoleucine), which are metabolized in skeletal muscle [52]. This is the reason for the higher levels of alanine and glutamine and the lower levels of branched chain amino acids in the 5% and 10% groups. It is well known that higher plasma glutamine levels can inhibit skeletal muscle catabolism and maintain lean tissue [53,54]. These results suggest that high-carbohydrate low-fat nutritional support is beneficial in reducing skeletal muscle catabolism and maintaining plasma amino acid stability after burn injury. In the two low-fat groups, the 10% fat group was more effective when combined with trends in REE, body weight and plasma protein levels, suggesting that although a lower fat supply could moderately reduce burn-mediated hypermetabolic responses and mitigate protein catabolism, it is not “the lower, the better”.

In this study, we found that different carbohydrate–fat ratios of nutrition formulations can affect insulin levels and glucose tolerance in burned rats. After 14 days of continuous administration of different nutritional supports, insulin levels in the 5% and 10% fat groups were significantly higher than those in the 20% and 30% fat groups, indicating that supplementation with higher carbohydrate levels can stimulate insulin secretion effectively, which is important for maintaining blood glucose stability. Several clinical studies have confirmed that patients with high carbohydrate nutritional support have significantly higher serum insulin levels and are closely related to the nitrogen-sparing effect of glucose [30,47].The results of the glucose tolerance experiment showed that the glucose tolerance level of the 10% fat group was better than that of the other groups at 7–14 days after burns, especially at 10–14 days, and the mechanism may be related to the lower skeletal muscle loss and higher insulin level. However, glucose tolerance levels in the 5% fat group were not superior to those in the 20% and 30% fat groups, suggesting that relatively lower fat intake is beneficial for improving glucose metabolism after burns, but excessively lower fat intake is not optimal. This conclusion is supported by data on organ damage. The major organs (heart, liver and kidney) were remarkably damaged post-burn, and all serum enzymatic indices reflecting organ damage substantially increased. In contrast, the degree of organ damage in the 10% fat group was obviously lower than that in the other three groups, especially the liver function damaged slightly, which is the pathophysiological basis for the more stable plasma amino acid and protein levels in rats in this group. The experimental results suggest that lower fat intake is beneficial for maintaining hormone levels and basic stability of blood glucose and reducing the degree of organ damage after burns. However, the lower the fat supply is not better; for burned rats, the proportion of fat should not be lower than 10%.

In this study, we found that the optimal nutrient ratio for burned rats, i.e., the ratio of protein, carbohydrate and fat, was 20%:70%:10%, which provides evidence for the optimal nutritional formulation for future animal experiments on burn nutrition. In our previous animal experiments, the proportion of fat was approximately 20–30%, which is a common formula in burn patients [53,55]. As metabolism is an adaptive issue, the structure of the diet and the ratio of nutrients greatly influence the metabolic pattern of rats. The clarification in this research that the 10% fat group is more effective in metabolic support in burned rats is only for the results obtained in rats. Typically, the fat content of rat chow is about 4%, so it is appropriate to give a fat ratio of 10%. When the fat content is 10%, it not only meets the range of the diet structure of rats, but also enables the body to obtain the required nutritional support. For this reason, we believe that only a rationing scheme that forms a specific proportional relationship with the fat content of normal foods will acceptable to the organism, and that the fat content should not be too low or too high. In the case of burn patients, due to differences in dietary background, the nutritional rationing guidelines of Western countries are not suitable for Asian patients. Consequently, the nutrition supplementation should take full consideration of their dietary structure and develop a nutritional rationing program. This study found that there are great differences in the demand for fat between animals and humans, which may be related to the differences in their dietary structures. However, the optimal proportion of nonprotein calories in burn patients needs to be explored by multicenter randomized controlled clinical tests. In addition, limited by animal experiments, this study only observed the effects of different carbohydrate–fat ratios in enteral nutrition on metabolic changes and organ damage, but did not consider the effects of parenteral nutrition. In fact, different routes of nutritional support have obvious effects on body metabolism, especially excessive intravenous glucose supplementation, which can lead to a surge in blood glucose levels which is detrimental to patients’ glycemic control [47,56]. This is the deficiency in parenternal nutrition with high-glucose low-fat, which should be given more attention in the nutrition therapy of patients with severe burns.

## 5. Conclusions

There is great controversy regarding the ratio of carbohydrates to fats in the nutritional treatment of burns. Therefore, this study used a burn animal model to explore the effects of different carbohydrate–fat supplies. We found that lower fat intake is beneficial to maintaining metabolic stability and lessening organ damage after burns, but it is not “the lower, the better”; the percentage of fat should not be lower than 10%. The finding provides an experimental basis for the optimization of nutritional formulations for burn patients.

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
