# Peer review of "Effects of Different Ratios of Carbohydrate–Fat in Enteral Nutrition on Metabolic Pattern and Organ Damage in Burned Rats"

_nutrients, 2022, doi:10.3390/nu14173653_

Round 1

Reviewer 1 Report

A good scientific study

Author Response

Thank you for your recognition of our research.

Reviewer 2 Report

This is a very nicely written and informative manuscript. The English is fine, only minor typos need correction. The manuscript is structured and easy to follow, the methods are described clearly. The figures add to the appeal, but are overall "too much" and very colorful. I kindly suggest to reduce the visual overflow and seperate some panels as described below to facilitate reading and understanding your work.

Abstract: please add some details about duration of study and amount of nutrition provided, as well as about which outcome parameters were measured at which timepoints

This is a very nicely written and informative chapter

Line 47: I disagree that the amount of macronutrients that should be provided to a severely burned patient is agreed upon, please rephrase

Line 51: suggest to weaken this statement: “may be of importance”

Line 55: there is no international consensus on protein requirements – recommendations range from 0.8 to 2.5 g/kg/d between various guidelines. Please correct and/or address this issue.

Line 127: suggest to add a short sentence “as described in section…”

Line 251: is it valid to compare metabolite signals of humans with those of rats?

Line 284: Please separate the figures 1A to 1F so that every figure can have a short description. As it is now, the reader may not understand each figure. Please consider that not all readers of nutrients are experienced in reading these panels. The lack of titles on the axes and sheer amount of numbers provided adds to the overall confusing display.

Line 345 and 374 and 395: please explain the color code of figure 3F, figures 4 & 5, if the colors do not provide additional information other than the groups I suggest to remove it/ reduce to greyscale.

Discussion:

It seems unexpected that the 10% fat group performed best at all parameters. What were likely confounders?

Please add some strengths and limitations to your work.

Author Response

Dear Reviewers and Editors,

Thank you for your letter and the reviewers’ comments on our manuscript (ID: nutrients-1869135). The comments are very helpful for revising and improving our paper, and they’re of the important guiding significance to another research. After carefully studying the reviewer's comments, we have made corresponding amendments to the paper.

We completed the revisions exactly as requested by the reviewers and editors and used the "track changes" feature to make the changes, including: (1) We checked all references to ensure that they were closely related to the content of the paper. (2) We added descriptions of study duration, nutrient amounts, measurement time points, and outcome parameters to the abstract as requested by the reviewers. (3) We revised vague and non-specific descriptions in the article based on the reviewers' comments. (4) We corrected the color of Figures 3-5 based on the reviewers' comments. (5) We conducted a full-text check, focusing on repetition in the Materials and Methods section. (6) Based on the reviewer's comments, we added a discussion of the mechanism of the better effect of 10% fat in rats to the discussion.

Below is our response to each of the review comments.

Reviewer 2

1. Abstract: please add some details about duration of study and amount of nutrition provided, as well as about which outcome parameters were measured at which time points.

Reply: Thank you for your professional comments. As you suggested, we have added a description of the study duration, nutrient amounts, measurement time points and result parameters to the abstract (Detail is provided on Lines 25-30).

2. Line 47: I disagree that the amount of macronutrients that should be provided to a severely burned patient is agreed upon, please rephrase

Reply: Thank you very much for your valuable comment. Following your suggestion, we have revised this description (Detail is provided on Lines 47).

3. Line 51: suggest to weaken this statement: “may be of importance”

Reply: Thank you very much for reminding. We have revised these representations (Detail is provided on Lines 54).

4. Line 55: there is no international consensus on protein requirements – recommendations range from 0.8 to 2.5 g/kg/d between various guidelines. Please correct and/or address this issue.

Reply: Thank you for your professional and precise questions. Based on your suggestion, we have corrected the recommended serving range of protein (Detail is provided on Lines 58).

 5. Line 127: suggest to add a short sentence “as described in section…”

Reply: As you suggested, we add a short sentence "as described in section 2.3.2" on line 127

6. Line 251: is it valid to compare metabolite signals of humans with those of rats?

Reply: The NMR signal signatures of human plasma metabolites differ from those of rat plasma and are not suitable for direct intercomparison, but for one specific compound, HMDB data can be useful for compound identification, even if the same compound from different species sources still exhibits similar signatures. There is some similar literature that uses this approach, such as:

[1] Liao S, Li P, Wang J, Zhang Q, Xu D, Yang M, Kong L. Protection of baicalin against lipopolysaccharide induced liver and kidney injuries based on 1H NMR metabolomic profiling. Toxicol Res (Camb). 2016 May 24;5(4):1148-1159.

[2] Deborde C, Hounoum BM, Moing A, Maucourt M, Jacob D, Corraze G, Médale F, Fauconneau B. Putative imbalanced amino acid metabolism in rainbow trout long term fed a plant-based diet as revealed by 1H-NMR metabolomics. J Nutr Sci. 2021 Feb 24;10:e13.

7. Line 284: Please separate the figures 1A to 1F so that every figure can have a short description. As it is now, the reader may not understand each figure. Please consider that not all readers of nutrients are experienced in reading these panels. The lack of titles on the axes and sheer amount of numbers provided adds to the overall confusing display.

Reply: As you suggested, we have added graphical description and the definition of the coordinate axes (Detail is provided on Lines 284)..

8. Line 345 and 374 and 395: please explain the color code of figure 3F, figures 4 & 5, if the colors do not provide additional information other than the groups I suggest to remove it/ reduce to greyscale.

Reply: Following your suggestion, we have modified the color of Figure 3F, Figure 4 and Figure 5 to a single color. (Detail is provided on Lines356, 383, 406).

9. It seems unexpected that the 10% fat group performed best at all parameters. What were likely confounders? Please add some strengths and limitations to your work.

Reply: Thank you for your professional comments. We modified the Discussion section to address the possible mechanism of the effects of the 10% fat group on burnt rats (Detail is provided on Lines 514-527).

Reviewer 3 Report

The manuscript, entitled “Effects of different ratios of carbohydrate-fat in enteral nutrition on metabolic pattern and organ damage in burned rats”, tried to get the best ratio of carbohydrate-fat in the recovery of burn-induced disruptions of metabolic pattern and organ damage. It’s remarkably interesting project and provide a good guideline for nutrition therapy after burn injury, which is that lower fat intake is beneficial to maintain metabolic stability and improve organ damage after burns, however, the percentage of fat supply should not be less than 10% in burned rats. I have some regrets by reading thoroughly, the greatest regret is that the organ damage markers were questionable. It’s common that burn-induced heart dysfunction happened in the early and acute pages (till post 48 hours post burn and then completely recovery until getting sepsis by measurement of ECHO). The data presented in this manuscript were contrary and contradictory, please explain the errors. In addition, I also have below concerns and hope the revised version can be addressed them.

1.       There was 40% of plagiarism by checking it in www.ithenticate.com.

2.       To protect blood cells integrity, the centrifuge speed should be lower than 3000 g for 15-30 mins. Authors applied 16,000 rpm for 10 mins and should interpret the method, otherwise, the plasma contained lots of components from blood cells.

3.       It’s better to understand the results if author provides the HMR spectra curves containing each group in Fig. 1A.

4.       Strongly suggest that author expect the potential mechanisms of why 10% group is best beneficial for burn injury patients in discussion part.

5.       Do you have histological data for organs? Do you determine the biomarkers using tissues?

Author Response

Dear Reviewers and Editors,

Thank you for your letter and the reviewers’ comments on our manuscript (ID: nutrients-1869135). The comments are very helpful for revising and improving our paper, and they’re of the important guiding significance to another research. After carefully studying the reviewer's comments, we have made corresponding amendments to the paper.

We completed the revisions exactly as requested by the reviewers and editors and used the "track changes" feature to make the changes, including: (1) We checked all references to ensure that they were closely related to the content of the paper. (2) We added descriptions of study duration, nutrient amounts, measurement time points, and outcome parameters to the abstract as requested by the reviewers. (3) We revised vague and non-specific descriptions in the article based on the reviewers' comments. (4) We corrected the color of Figures 3-5 based on the reviewers' comments. (5) We conducted a full-text check, focusing on repetition in the Materials and Methods section. (6) Based on the reviewer's comments, we added a discussion of the mechanism of the better effect of 10% fat in rats to the discussion.

Below is our response to each of the review comments.

Reviewer 3:

1. The manuscript, entitled “Effects of different ratios of carbohydrate-fat in enteral nutrition on metabolic pattern and organ damage in burned rats”, tried to get the best ratio of carbohydrate-fat in the recovery of burn-induced disruptions of metabolic pattern and organ damage. It’s remarkably interesting project and provide a good guideline for nutrition therapy after burn injury, which is that lower fat intake is beneficial to maintain metabolic stability and improve organ damage after burns, however, the percentage of fat supply should not be less than 10% in burned rats. I have some regrets by reading thoroughly, the greatest regret is that the organ damage markers were questionable. It’s common that burn-induced heart dysfunction happened in the early and acute pages (till post 48 hours post burn and then completely recovery until getting sepsis by measurement of ECHO). The data presented in this manuscript were contrary and contradictory, please explain the errors. In addition, I also have below concerns and hope the revised version can be addressed them.

Reply: We apologize for the misunderstanding caused by our descriptive errors. We strongly agree with your point of view. In general, severe burn trauma induces a unique complex of responses in the body that can be divided into two parts [1, 2]. The first phase ("ebb phase") lasts 48-96 hours and leads to a hypometabolic response in the body, resulting in a decrease in cardiac output and oxygen consumption [1]. This is followed by a rapid shift to a second phase ("flow phase") in which the maximum peak is usually 5 days after the trauma and can persist for years after the initial injury [2-4]. The aim of this study was to evaluate the effect of different carbohydrate-fat ratios on the metabolic pattern of rats, while the effect of nutrition on metabolism tends to take a longer time, so we focused on the changes in the second phase.. To avoid misunderstandings by readers, we modified the description of cardiac dysfunction to myocardial cell injury (Detail is provided on Lines 184, 398).

[1].   Auger, C, Osai S, Marc GJ. The biochemical alterations underlying post-burn hypermetabolism. Biochim Biophys Acta Mol Basis Dis, 2017. 1863(10 Pt B): p. 2633-2644.

[2].   Guillory, AN, Clayton, RP, Herndon, DN, Finnerty, CC. Cardiovascular Dysfunction Following Burn Injury: What We Have Learned from Rat and Mouse Models. Int J Mol Sci, 2016. 17(1).

[3].   Jeschke MG, Gauglitz GG, Kulp GA, Finnerty, CC, Williams, FN, Kraft, R, Suman, OE, Mlcak, RP, Herndon, DN, Long-term persistance of the pathophysiologic response to severe burn injury. PLoS One, 2011. 6(7): p. e21245.

[4].   Krbcova MV, Zajicek R, Bednar F. Burn-Induced Cardiac Dysfunction: A Brief Review and Long-Term Consequences for Cardiologists in Clinical Practice. Heart Lung Circ, 2021. 30(12): p. 1829-1833.

2. There was 40% of plagiarism by checking it in www.ithenticate.com.

Reply: Thank you for your suggestion. After we checked the full text and focused on changing the duplicated parts (mainly present in the material methods section).

3. To protect blood cells integrity, the centrifuge speed should be lower than 3000 g for 15-30 mins. Authors applied 16,000 rpm for 10 mins and should interpret the method, otherwise, the plasma contained lots of components from blood cells.

Reply: We apologize for the error in the material methods, in fact the subject of the 16,000 rpm centrifugation for 10 minutes was the plasma frozen after centrifugation in section 2.3.4. We have corrected this error in the text (Detail is provided on Lines 204).

 4. It’s better to understand the results if author provides the HMR spectra curves containing each group in Fig. 1A.

Reply: Thank you for your insightful advice. Based on your suggestions, we have added these HMR spectra curves. Due to the size limitation of Figure 1, the HMR spectra curves from other groups were laid out in new Fig. S1.

 5. Strongly suggest that author expect the potential mechanisms of why 10% group is best beneficial for burn injury patients in discussion part.

Reply: Thank you for your professional and precise questions. We have added a section to the Discussion section aimed at illustrating the mechanisms by which the 10% fat group may be beneficial in burned rats (Detail is provided on Lines 514-527).

6. Do you have histological data for organs? Do you determine the biomarkers using tissues?

Reply: Thanks for the reminder, unfortunately we did not have access to histological data of organs or to data identifying tissue damage using biomarkers. At present, we only have data from peripheral blood reflecting markers of tissue damage or cellular damage, but these data also reflect conditions associated with tissue or cellular damage in the organism, which is an important manifestation of organismal damage. In subsequent articles, we will supplement the data with relevant data from tissues to ensure that the data are adequate and complete.

Round 2

Reviewer 3 Report

I am happy to see the revised cersion and do think that authors are serious to pay attention my concerns. The responsed addressed my questions. So, I agree to be going to publish it as curretn version.

Thanks!

Author Response

Thank you for your professional comments